# Bayesian inference of admixture graphs on Native American and Arctic populations

Svend V. Nielsen[1]☯, Andrew H. Vaughn [2]☯*, Kalle Leppälä[1,3], Michael J. Landis[4], Thomas Mailund[1], Rasmus Nielsen[5,6]

**1** Bioinformatics Research Centre, Aarhus University, Aarhus, Denmark, **2** Center for Computational Biology, University of California Berkeley, Berkeley, California, United States of America, **3** Research Unit of Mathematical Sciences, University of Oulu, Oulu, Finland, **4** Department of Biology, Washington University in St. Louis, St. Louis, Missouri, United States of America, **5** Departments of Integrative Biology and Statistics, University of California Berkeley, Berkeley, California, United States of America, **6** Center for GeoGenetics, University of Copenhagen, Copenhagen, Denmark

☯ These authors contributed equally to this work.
* ahv36@berkeley.edu

**Data Availability Statement:** All data are fully available without restriction at https://github.com/avaughn271/AdmixtureBayes.

## Abstract

Admixture graphs are mathematical structures that describe the ancestry of populations in terms of divergence and merging (admixing) of ancestral populations as a graph. An admixture graph consists of a graph topology, branch lengths, and admixture proportions. The branch lengths and admixture proportions can be estimated using numerous numerical optimization methods, but inferring the topology involves a combinatorial search for which no polynomial algorithm is known. In this paper, we present a reversible jump MCMC algorithm for sampling high-probability admixture graphs and show that this approach works well both as a heuristic search for a single best-fitting graph and for summarizing shared features extracted from posterior samples of graphs. We apply the method to 11 Native American and Siberian populations and exploit the shared structure of high-probability graphs to characterize the relationship between Saqqaq, Inuit, Koryaks, and Athabascans. Our analyses show that the Saqqaq is not a good proxy for the previously identified gene flow from Arctic people into the Na-Dene speaking Athabascans.

## Author summary

One way of summarizing historical relationships between genetic samples is by constructing an admixture graph. An admixture graph describes the demographic history of a set of populations as a directed acyclic graph representing population splits and mergers. The greedy search algorithms that are typically used to infer admixture graphs may fail to find the globally optimal graph. We here improve on these approaches by developing a novel MCMC sampling method, *AdmixtureBayes*, that can sample from the posterior distribution of admixture graphs. This enables an effective search of the entire state space as well as the ability to report a level of confidence in the sampled graphs. We apply Admixture-Bayes to a set of Native American and Arctic genomes to reconstruct the demographic history of these populations and report posterior probabilities of specific admixture

**Funding:** RN received grant R01GM138634, National Institutes of Health https://www.nih.gov. AHV received grant 2146752, National Science Foundation Graduate Research Fellowship Program https://www.nsfgrfp.org. The funders had no role in study design, data collection and analysis, decision to publish, or preparation of the manuscript.

**Competing interests:** The authors have declared that no competing interests exist.

events. While some previous studies have identified the ancient Saqqaq culture as a source of introgression into Athabascans, we instead find that it is the Siberian Koryak population, not the Saqqaq, that serves as the best proxy for gene flow into Athabascans.

## Introduction

Admixture graphs [1] provide a concise description of the historical demographic relationships between genetic samples of populations, assuming their relationships are the product of discrete, instantaneous splits and admixture events. The assumption of discrete, instantaneous events is clearly an oversimplification for most real data, but it facilitates interpretation and makes admixture graphs a popular first step in analyses. Each graph topology is associated with parameters capturing genetic drift and admixture proportions, and once these are fitted to genetic data, the goodness of fit can be measured to determine how accurately the graph captures the historical relationship between samples. Inferring graph topologies, however, involves a combinatorial search, and since the space of graphs grows super-exponentially in the number of populations and the number of admixture events, an exhaustive search is typically not possible. Instead, the search for well-fitting topologies is often done manually or through greedy algorithms.

The most popular methods for estimating admixture graphs are *TreeMix* by Pickrell and Pritchard [2], *qpGraph* by Patterson *et al.* [1], and *OrientAGraph* by Molloy *et al.* [3], all of which take a greedy approach to searching the state space of graph topologies. qpGraph allows users to sequentially identify the best phylogenetic position of a possibly admixed population in a previously established admixture graph and evaluate the improved fit in terms of simple allele-sharing statistics. The program *MixMapper* by Lipson *et al.* [4] employs a similar strategy and has options for fitting up to two admixture events simultanously. TreeMix estimates an admixture graph *de novo* by automatically estimating the best tree without admixture events followed by automatic, sequential insertion of the admixture branches. In contrast to MixMapper and qpGraph, TreeMix searches through potential admixture graphs without user input by way of an efficient greedy heuristic. OrientAGraph is based on the TreeMix approach, but adds a technique called maximum likelihood network orientation, which helps avoid getting stuck in incorrect local optima during the optimization process. The method *miqograph* employs mixed-integer quadratic optimization to search through the space of admixture graphs but restricts admixture events to the leaf nodes of the graph [5].

To penalize deviations from the expected and observed allele sharing statistics, all five methods use a Gaussian model for the distribution of allele frequencies among populations. The implicit assumption in the Gaussian model is that changes in allele frequency due to genetic drift can be approximated as a Brownian motion process. This assumption dates back to the early work by Edwards and Cavalli-Sforza [6] and has recently re-emerged as a computationally attractive alternative to the full Wright-Fisher process. It has previously been used in several other methods aimed at modeling the joint distribution of allele frequencies among populations [7] [8] [9].

There are also phylogenetic network methods that infer admixture graphs using sets of locus-specific gene trees as nuisance parameters which are either pre-estimated [10] or integrated out using MCMC [11] [12]. These approaches must evaluate the likelihood of each gene tree separately, making them more computationally expensive and therefore limited to fewer populations than the Gaussian drift models. To handle larger datasets, some methods summarize all the gene trees into a few statistics that are evaluated with a pseudolikelihood [13] [14]

for a small reduction in accuracy [14]. In terms of speed, these pseudolikelihood methods are similar to the Gaussian drift methods. However, the Gaussian approach offers a way to instead approximate a true likelihood (rather than a pseudolikelihood), which we use in this paper.

The greedy search algorithms used by current methods do not guarantee that the inferred graph is optimal. In practice, the optimal graph found by a greedy search can potentially be very different from better-fitting, but never-discovered, graphs [3] [15]. Regardless of whether a search finds the optimal graph or not, if a single graph is inferred and used for all downstream analysis, that point estimate would not intrinsically report confidence in various estimated features, such as the topology of relationships among populations, the presence or absence of admixture events, and the intensity of those events. There might be many graphs that fit the data equally well, and we should have more confidence in features shared among many of them than we should in features that are only found in some of them; shared features are most likely signals in the data while those that rarely occur are most likely spurious. Analyses based on a single graph do not distinguish between features that are estimated with high confidence and those estimated with low confidence. While it is possible to generate a distribution of TreeMix graphs across independent analyses of bootstrap replicates, it is rarely done in practice.

Here, we provide an alternative to greedy searches. Based on a model similar to TreeMix and qpGraph, we develop a Bayesian approach to sample over the graph-space using a reversible jump Markov Chain Monte Carlo (MCMC) method. The method can identify the graph with the highest likelihood encountered by the MCMC sampler, thereby effectively working as a heuristic maximum-likelihood optimizer, or it can report several summaries of the posterior distribution of admixture graphs. For example, it can estimate the graph topology with the highest marginal posterior when integrated over admixture and divergence times as measured by occupancy in the MCMC sampler. A marginal posterior is computed in *admixturegraph* [16] as well, but the exhaustive search algorithm of admixturegraph finds the graph with the highest posterior—not the graph shape with the highest marginal posterior. A particular strength of our new method is that it circumvents the need to report a single best graph by allowing calculations of posterior probabilities of particular marginal relationships between populations. We consider three approaches for this: one based on simplifying admixture graphs into simpler structures, one based on summarizing shared topologies into a consensus graph, and one based on subgraph analysis. If the number of leaves in the considered subgraph is kept small, we will observe few distinct subgraphs with these leaves, and we can estimate a complete posterior distribution over these graphs. Sampling subgraphs from the space of full graphs allows us to incorporate information from other populations when exploring the relationship between a subset of the populations.

We illustrate the utility of our method using simulations and reanalyze a previously published genomic dataset of Siberians and Native Americans [17]. We use the method to revisit two important and controversial questions in the history of the peopling of the Americas. First, we analyze the origin of the Inuit and show that they are modeled best as an admixture between a population represented by the Saqqaq genome, and a population represented by Athabascans. Secondly, we show that Athabascans are best represented as admixed between a Native American population and a Siberian population most closely related to the Koryak, but not the Saqqaq.

## Results

### Method overview

We here present *AdmixtureBayes*, an MCMC algorithm for sampling admixture graphs from their posterior distribution, given a set of genetic data from multiple populations.

We begin by presenting our formal definition of an admixture graph. An admixture graph consists of a topology and a set of continuous parameters. The space of topologies for a given number of leaves, $L$, consists of all uniquely labeled graphs of the set of all directed acyclic graphs which fulfill

1. There exists one and only one root. That is a node with one parent (the outgroup) and exactly two children.

2. The number of nodes of degree 1 is $L + 1$. $L$ of these nodes have only one parent and are called leaves. 1 of these nodes is called the outgroup and has exactly one child, the root.

3. If a node is not the root, a leaf, or the outgroup, it has either

    a. 1 parent and 2 children in which case we call it a divergence node.

    b. 2 parents and 1 child in which case we call it an admixture node.

4. There are no *eyes*, i.e. the parent nodes of an admixture node are distinct (and the child nodes of a divergence node are distinct).

    The labeling consists of

1. All leaves and the outgroup are given a unique label.

2. Parent edges of an admixture node can be either a 'main' branch or an 'admixture' branch. All admixture nodes have one parent edge of each type.

We do not label branches and nodes in general, meaning that even though the the leaves are given a unique label, the leaves themselves are not unique. For example, switching the labels of two leaves that form a cherry in the graph, would not change the graph topology. For a more formal definition, see the definition of topology in S1 Text. All branches have a length in the interval $(0, \infty)$ and all admixture nodes are given an admixture proportion in the interval $(0, 1)$.

The Methods section describes our implementation of a Markov Chain Monte Carlo (MCMC) algorithm, AdmixtureBayes, which samples admixture graphs from their posterior distribution. We summarize genetic data from multiple populations as a matrix that captures how allele frequencies in the data covary between populations. AdmixtureBayes samples graphs that explain this covariance matrix. The topology of any sampled graph captures the relationships between samples as a mixture of the graphically structured covariance matrices. Branch lengths capture the amount of genetic divergence between populations, measured by drift, and admixture events explain shared allelic covariance between otherwise independently evolving populations. As a property of the MCMC algorithm, each graph is sampled at a frequency corresponding to its posterior probability. AdmixtureBayes is available to use at https://github.com/avaughn271/AdmixtureBayes.

## Comparisons with TreeMix and OrientAGraph

We compared the accuracy of AdmixtureBayes to TreeMix and OrientAGraph on 4 distinct admixture graphs, shown in Fig 1. We simulated datasets from each of these admixture graphs in *msprime* [18] by using the `add_population_split` and `add_admixture` options and adjusting event times and population sizes until the allele frequency drift terms matched those of the admixture graph.

We then analyzed all simulated datasets with AdmixtureBayes, TreeMix, and OrientAGraph (see the section "Running AdmixtureBayes, TreeMix, and OrientAGraph" for details). Comparing their accuracy is not straightforward because TreeMix and OrientAGraph produce

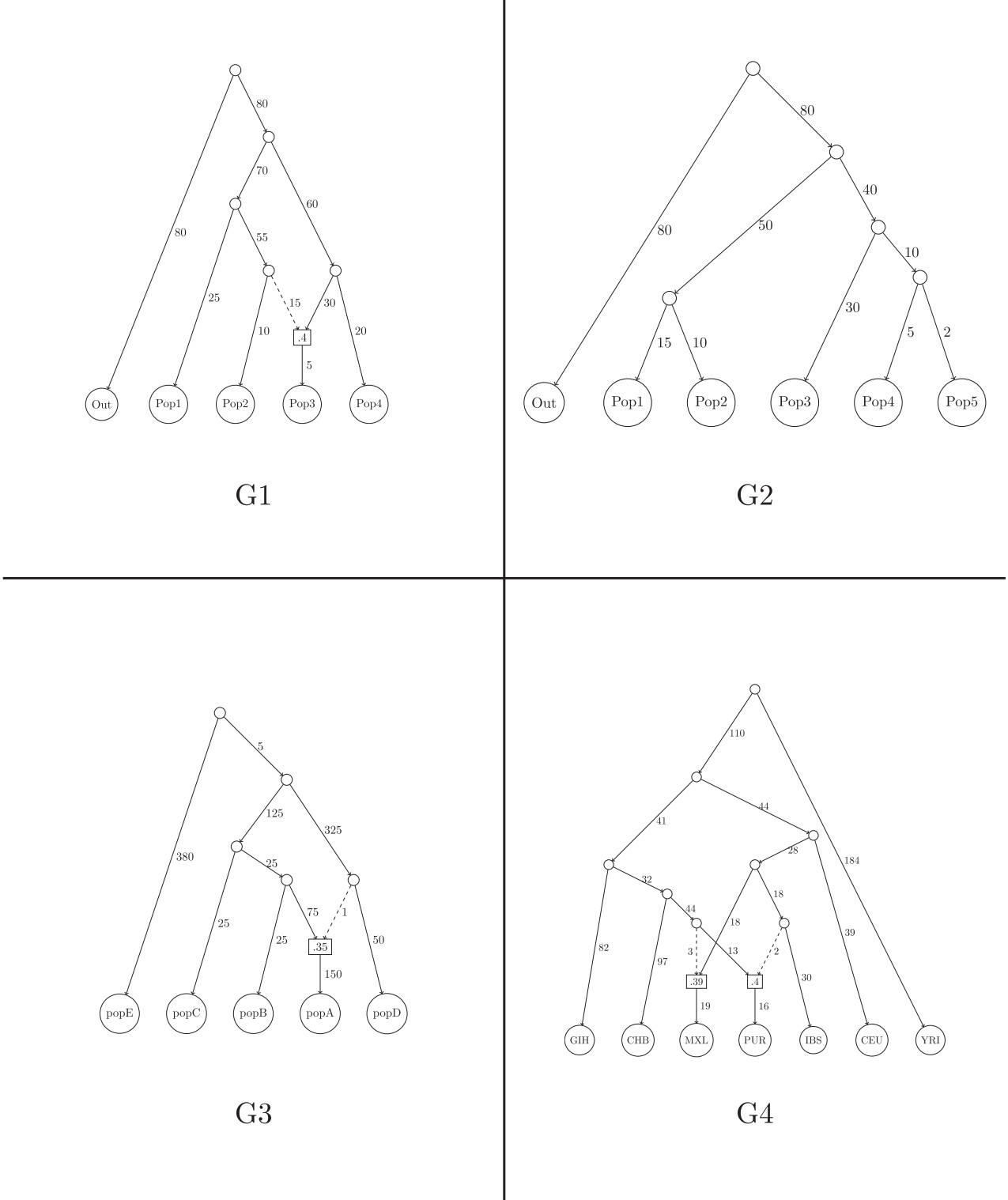

**Fig 1. The graphs G1, G2, G3, and G4 used for the comparisons between methods.** G1 and G2 are not based on any real dataset, but the branch lengths are chosen to have human-like values. Out was used as the outgroup for both graphs. G3 is based on M1 from Molloy *et al.* (2021), the graph that motivated the development of the MLNO approach of OrientAGraph. We have changed some of the branch lengths. popE was used as the outgroup. G4 is based on Model M7 from Fig 3 of Molloy *et al.* (2021), which is in turn based on Fig 7a from Wu (2020) [19]. The populations ITU, JPT, and ASW have been removed. The YRI population was used as the outgroup. For all graphs, as in Molloy *et al.* (2021), branch lengths are not shown to scale and are shown multiplied by 1000. Divergence nodes are shown as circles. Admixture nodes are shown as rectangles. The fractions inside the admixture nodes denote the contribution from the population represented by the dashed line.

one graph whereas AdmixtureBayes produces posterior samples of graphs. In addition, Tree-Mix and OrientAGraph assume a fixed number of admixture events, whereas AdmixtureBayes samples graphs with different numbers of admixture events. We ran TreeMix and OrientA-Graph conditioned on the true number of admixture events, while we considered all graphs produced by AdmixtureBayes, even those with the wrong number of admixture events. We note that this could increase the error of AdmixtureBayes. Furthermore, both TreeMix and OrientAGraph allow admixture involving the branch to the outgroup, which AdmixtureBayes does not. The extent to which this was a problem varied between simulation models, so we handled this on a case-by-case basis. We used three metrics to compare the graphs inferred by these methods to the true underlying admixture graph. The Topology Equality is a simple metric that is 1 if the inferred graph has the same topology as the true graph and 0 otherwise. The next metric we considered is the Covariance Distance, defined as the Frobenius distance between the allelic covariance matrix of the true graph and the allelic covariance matrix of the inferred graph (see Methods). Finally, we measured the Set Distance, which we defined as a topological distance measure similar to the Robinson-Foulds metric (S9 Fig; Methods section).

For each of the 4 admixture graphs we analyzed (see Fig 1), we performed the following analysis: 20 independent datasets were simulated using msprime and all three methods were run on each dataset. Then, each of the three metrics was calculated for the results of each method. For AdmixtureBayes, we measured both the accuracy of the sampled graph with the highest posterior (we call this the AdmixtureBayes Mode) and the mean accuracy of a graph sampled from the posterior (we call this the AdmixtureBayes Mean). We plot the values of these metrics across the 20 datasets as boxplots in Fig 2. We also highlight that an excellent comparison of TreeMix, OrientAGraph, and miqograph was done in Molloy *et al.* [3], which both illustrated OrientAGraph's ability to infer topologies TreeMix could not and demonstrated that miqograph was unable to infer topologies with deep admixture events.

On graph G1, which contains 1 admixture event, all methods perform similarly well. The correct topology was inferred by all methods on all datasets (giving a Set Distance value of 0), and the accuracy of the covariance matrix implied by each of the inferred graphs (as measured by the Covariance Distance) is quite similar.

On graph G2, which contains no admixture events, TreeMix and OrientAGraph are able to infer the correct topology for all 20 datasets. The Mode estimate of AdmixtureBayes also infers the correct topology in all cases. For all datasets, the AdmixtureBayes Mean topologies are highly concentrated on the true topology, though there is some variation. This is to be expected given the inherent noise in the data. It is also worth noting that the incorrectly inferred topologies sampled by AdmixtureBayes may include graphs with an admixture event, an error which we do not allow TreeMix and OrientAGraph to make as we run them with the correct number of admixture events (zero). We note that the AdmixtureBayes Covariance Distance is slightly larger than the TreeMix and OrientAGraph distances. This is to be expected as both of those methods explicitly perform optimization on branch lengths and admixture proportions, which will likely result in a better model fit than the graph AdmixtureBayes samples that happens to have the highest posterior.

On graph G3, which has one admixture event, TreeMix does quite poorly. This is by design, as G3 is based on Model M1 from Molloy *et al.* [3], which motivated the development of OrientAGraph. In particular, TreeMix incorrectly infers an admixure event involving the outgroup in 17 out of the 20 datasets. Of the 3 remaining datasets, TreeMix was only able to infer the correct topology for 2 of them. We only plot the accuracy statistics for the 3 graphs that do not involve admixture with the outgroup as these are the only graphs that exist in the same state space as AdmixtureBayes. However, we highlight that the boxplots in Fig 2 do not necessarily represent all simulated datasets.

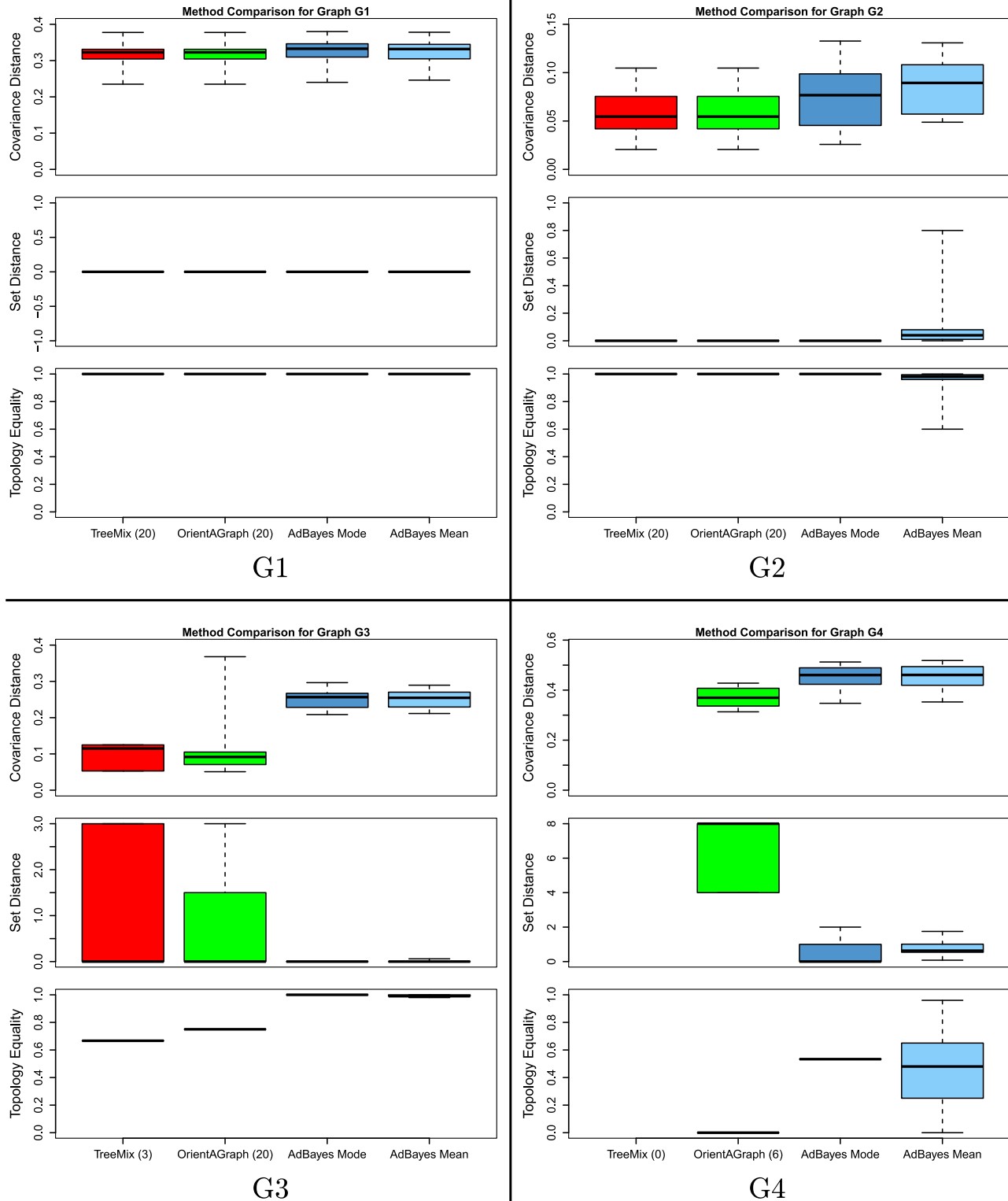

**Fig 2. We here plot the results of our method comparison with TreeMix and OrientAGraph.** For each of the graphs in Fig 1, we simulated 20 datasets and ran each method on each dataset. We compared the accuracy of each method with the 3 statistics discussed in the section Comparisons with TreeMix and OrientAGraph. For AdmixtureBayes, we examined both the Mode graph (the sampled graph with the highest posterior) and the mean value of the statistics when 100 graphs are sampled from the posterior (we refer to this as the AdmixtureBayes Mean). TreeMix and OrientAGraph allow admixture involving the outgroup, an error which AdmixtureBayes is not allowed to make. For fairness, we only plot the results for the graphs not

involving admixture with the outgroup. We have listed the number of datasets that resulted in such graphs in parentheses next to the method name on the x-axes. The Topology Equality statistic for TreeMix, OrientAGraph, and the AdmixtureBayes Mode can only be 0 or 1, so we plot a horizontal line at the mean value over the datasets, rather than a true boxplot.

In contrast to TreeMix, OrientAGraph never infers admixture involving the outgroup and infers the correct topology in almost 80% of all datasets. AdmixtureBayes, however, outperforms both methods by inferring the correct topology for all datasets, both using the Mode estimate and the Mean estimate. We attribute this to a superior framework for exploring the state space of topologies. We still note that TreeMix and OrientAGraph provide better estimates of branch lengths and admixture proportions, which we again attribute to the fact that AdmixtureBayes is not designed for optimizing the likelihood function for branch lengths but instead provides posterior distributions. If point estimates for branch lengths are of interest, we recommend that users optimize the branch lengths using other methods with the AdmixtureBayes Mode topology fixed.

Graph G4 represents a very complicated topology and is based on a model used by Molloy *et al.* [3] to represent the shortcomings of OrientAGraph. TreeMix incorrectly infers an admixture involving the outgroup for all datasets, so we do not plot the results from running TreeMix. OrientAGraph incorrectly infers an admixture involving the outgroup for 14 datasets, leaving 6 datasets to compare with AdmixtureBayes. We see that OrientAGraph never infers the correct topology and never has a Set Distance of less than 4. In contrast, the AdmixtureBayes Mode estimate represents the correct topology for more than half of all datasets, which we again attribute to a superior framework for exploring the state space of topologies. The AdmixtureBayes Mean estimates are fairly noisy, but still represent a posterior distribution that is often concentrated on the true topology. The optimization employed by OrientAGraph results in a lower Covariance Distance than AdmixtureBayes, even in the presence of an incorrect topology. Performing a similar optimization on the AdmixtureBayes Mode topology will likely yield a smaller Covariance Distance if a point estimate of an admixture graph with branch lengths is desired. From these results, we conclude that the MCMC framework of AdmixtureBayes provides an effective algorithm for searching through the topology space of admixture graphs and often infers the correct topology when other methods do not. All of the scripts used to run these simulations can be found in the SimulationStudy folder on the AdmixtureBayes GitHub.

### Exploring the genetic history of Saqqaq, Inuit and Native Americans

In the simulation section above, we demonstrated that AdmixtureBayes includes an effective algorithm for exploring the space of admixture graphs. However, the real advantage of the method is in its ability to quantify probabilities of graphs and subgraphs, and thereby to provide measures of statistical uncertainty. We illustrate the utility of the method on a set of previously published Siberian and Native American samples [17] to explore the relationship between Siberian Chukotko-Kamchatcan speakers (Koryak), an ancient individual from the extinct Saqqaq culture (Saqqaq), Inuit-Yupik-Unangan speakers (Greenlandic Inuit), and Na-Dene speakers (Athabascan). The dataset also contained North and South Americans (Anzick, Aymara) and various other groups. We chose the Yoruba population as the outgroup. Running time of AdmixtureBayes was 50 hours in parallel on 32 cores.

To extract information from the posterior distribution of admixture graph topologies, we introduce two ways of summarizing relationships among sets of focal populations (for details, see Methods). Both are based on summarizing each sampled admixture graph in the posterior into a *topology set*, which is the set of all nodes labeled by their descendants. This discards

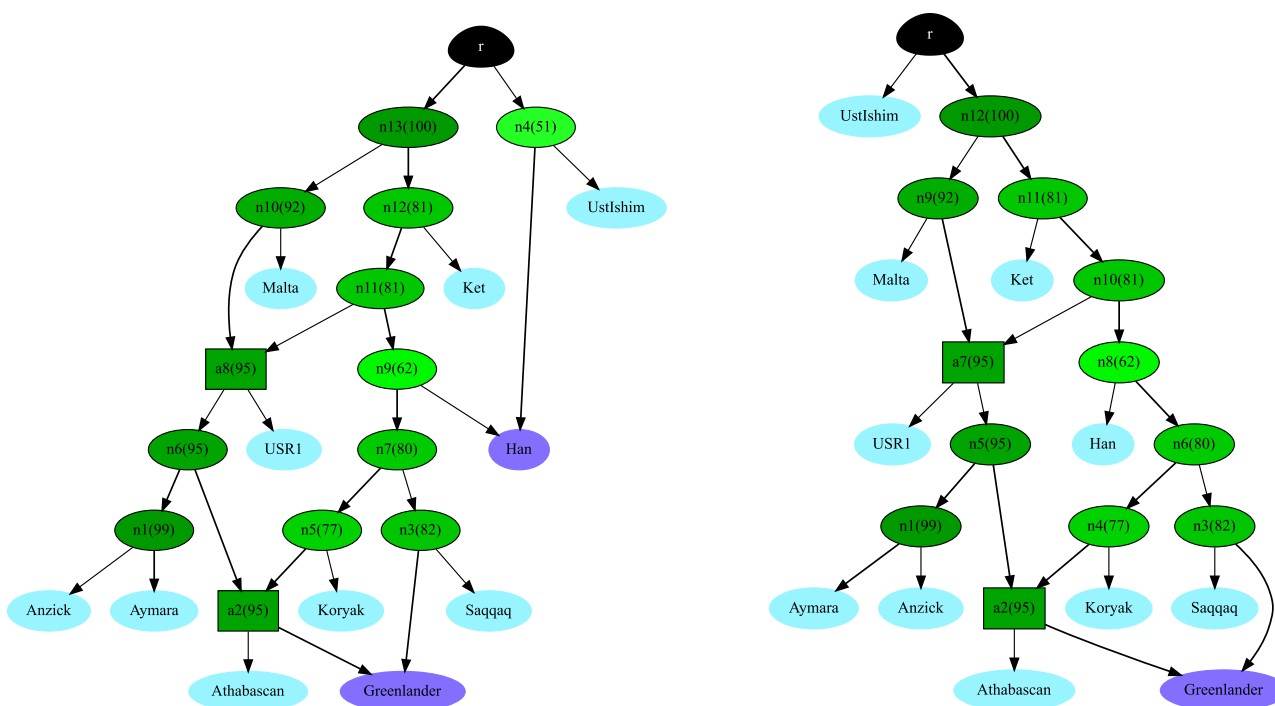

**Fig 3. The two minimal topologies with the highest posterior probabilities in our real dataset.** Leaf nodes that are the product of an admixture event are shown in purple. Leaf nodes that are not the product of an admixture event are shown in light blue. The root is shown in black. Each inner node is colored according to the posterior probability that the true graph has a node with the same descendants. Higher probabilities have a darker shade of green. The posterior probability is written as a percentage in parentheses inside each node, next to the node name, which is arbitrary. The left graph has a posterior probability of 32%. The right graph has a posterior probability of 19%.

information about the number of and timing of admixture events (see S9 Fig). From such a topology set, we can create the *minimal topology*, which is the 'simplest' directed graph yielding the same topology set (see S10 Fig). The two minimal topologies with the highest posterior probabilities are shown in Fig 3. We also considered the frequency of each internal node across posterior samples. In Fig 3 these frequencies are denoted as percentages in parentheses in each node. The second summary of the admixture graph sample is the set of nodes with a frequency higher than $\alpha$ in the topology sets, which we denote as the *consensus graph* at threshold $\alpha$. S6 Fig shows this summary for $\alpha = 0.75$.

While no single graph received high support when including all data, we can extract subgraphs that are informative about the relationships between specific subsets of populations. With AdmixtureBayes, it is possible to consider the relative support, in terms of posterior probability, of individual subgraphs. Analyzing the support for subgraphs within the context of a larger admixture graph has an advantage over analyses limited to the focal populations represented in the subgraph, that information from other populations can be directly taken into account.

There has been considerable debate about the relationships between populations represented by the Koryak, Saqqaq, Greenlanders, and the Athabascans. Archaeological evidence suggests that the Inuit people from Greenland and people from the now extinct Saqqaq culture represent independent migrations into the Americas from Eastern Siberia and the area around the Bering strait [20] [21] [22]. However, there is some debate about the origin of the Athabascans [21] [23] [24] [25]. Most molecular evidence of Athabascan ancestry is thought to have originated from the first migration of people into the Americas that also gave rise to most

other Native American groups, such as the indigenous people in Central and South America. However, some portion of genetic variation in Athabascans seems to have also originated from other groups, perhaps related to Inuit, Saqqaq, or other Siberians such as the Koryak. Naming and identifying sources of genetic variation is further complicated by the fact that these possible reference populations may themselves be admixed. A marginal analysis of the relationship between Koryak, Saqqaq, Greenlanders, and Athabascans, that can take gene flow from other groups into account, is therefore very much wanted.

S7 and S8 Figs depict the subgraphs for different subsets of these groups and for all groups together, extracted from the posterior distribution of graphs from the full dataset. The most strongly supported subgraph for Saqqaq, Athabascan, and Koryak supports the tree ((Athabascan, Koryak), Saqqaq) with 96% posterior probability. This implies that a relationship where the gene flow into Athabascans came from a population closer to the Saqqaq, than to the Koryak from Siberia, is not supported by the data. In contrast, when considering the relationship between Koryak, Athabascans and the Inuit Greenlanders, the most strongly supported admixture graph is a tree with the structure ((Athabascan, Greenlander), Koryak), likely reflecting gene flow into the Inuit Greenlander from Native Americans related to Athabascans.

We emphasize that in these inferences, by analyzing the posterior probability of subgraphs embedded within larger graphs, we have also explicitly modeled the effects of gene flow from other groups including various Siberian, Native American, and East Asian groups. When considering all four populations together, the Greenlanders are best modeled as a population admixed between Athabascan related populations and Saqqaq related populations. Again, there is no apparent gene flow between the Saqqaq and the Athabascans following their initial divergence. We also ran TreeMix and OrientAGraph on our dataset, each with a varying number of admixture events. We plot the results in S16 Fig. All of the scripts used to run the AdmixtureBayes analysis can be found in the RealDataAnalysis folder on the AdmixtureBayes GitHub, and the TreeMix and OrientAGraph results can be found in the OtherMethodsReal-Data folder.

## Discussion

We here present the program AdmixtureBayes, which is a method for inferring admixture graphs using MCMC. On simulated data, it infers graph topologies more accurately than both TreeMix and AdmixtureBayes, likely caused by these methods getting stuck in local optima topologies during the admixture edge addition process. However, we also note that even with AdmixtureBayes, the correct topology is not always inferred, suggesting that the reporting of a single "best graph" may not necessarily be best practice. As is common in phylogenetics, admixture graphs should report measures of statistical confidence for the relationships inferred among internal nodes in the graph, as is reported in this paper.

We also encourage the use of embedded subgraphs as a powerful approach for investigating the relationship between specific populations while taking gene flow from other reference populations into account, as was done in S7 and S8 Figs. The use of posterior probabilities, as reported here, is facilitated by the use of a bootstrap procedure that can estimate the effective number of independent SNPs. In our real data analysis, we obtained information from human genomes corresponding to approximately 40,000 independent SNPs. This number determines the peakedness of the likelihood surface, which directly influences the posterior distribution of admixture graphs. TreeMix and qpGraph employ similar resampling techniques to obtain variance estimates that control the peakedness of their likelihood surfaces.

Our analysis of Native American and Siberian samples largely recapitulates many previous analyses and identifies many admixture events [17]. Furthermore, we find a similar, but not

identical topology, to a previous admixture topology [17]. However, our results also indicate that several features of the true admixture graph remain uncertain. For example, we could not definitively resolve the question of introgression into the Han lineage from the ancestral lineage of Ust'-Ishim. Our analysis does not support previous claims that the Saqqaq culture is a good proxy for the source of gene flow into Athabascans [21] [24], although statistical power could still be improved.

In both our analysis and previous work, each population is represented by just one or two diploid individuals. Our simulations suggest that increasing the number of individuals per population might lead to substantially improved statistical accuracy (see S15 Fig). In addition, adding more populations, both modern and ancient, could change the results presented here, as there may be some ancestral components that are not adequately represented by the data we use in this analysis. We also note that the sample quality was relatively poor for some samples analyzed here, particularly the Saqqaq, which has many missing sites.

It is also worth noting that there are many other diverse fields such as linguistics, archaeology, and ethnography that seek to understand the historical relationships between different populations. While we do not incorporate data from these fields into this study, we do think that the results we present here are an important contribution towards clarifying the genetic evidence by improving on algorithms for admixture graph inference and correcting results that may have been caused by suboptimal optimization algorithms. However, we emphasize that the genetic evidence should be used in concert with other diverse fields in order to obtain an accurate picture of the historical movement and cultural development in the Arctic region and that the results in this paper should not, without further context, be used to infer cultural history.

The estimation of admixture graphs is becoming one of the most important tools in population genomics. However, methods for estimating such graphs are still in their infancy. AdmixtureBayes provides a step towards improved estimation and more rigorous quantification of statistical uncertainty in admixture graph inference.

## Methods

### AdmixtureBayes model

The AdmixtureBayes program searches the posterior distribution of admixture graphs given observed SNP data using a Markov Chain Monte Carlo procedure. To assess the likelihood of an admixture graph we summarize both the admixture graph and the data as covariance matrices of allele frequency changes [2]. The admixture graph covariance matrix is calculated as in TreeMix. Consider the tree structure in Fig 4 where population 2 is a mix of two ancestral populations with proportions $w$ and $1 - w$.

The allele frequency in the 4 populations, $P_0$, $P_1$, $P_2$ and $P_3$ are related through the allele frequency changes $x_0, \ldots, x_7$ at any SNP.

$$\begin{pmatrix} P_1 \\ P_2 \\ P_3 \end{pmatrix} - \begin{pmatrix} P_0 \\ P_0 \\ P_0 \end{pmatrix} = \begin{pmatrix} x_0 + x_1 + x_2 \\ x_0 + x_7 + w(x_6 + x_4) + (1 - w)(x_5 + x_2) \\ x_0 + x_3 + x_4 \end{pmatrix} = A \begin{pmatrix} x_0 \\ \vdots \\ x_7 \end{pmatrix}. \qquad (1)$$

Notice that $A$ is a matrix that only depends on the admixture graph through the graph structure and admixture proportions. We consider the vector of allele frequency drifts terms $(x_0 \cdots x_7)$ to be stochastic because it depends on a random sample of SNPs. In the neutral Wright-Fisher model, changes in allele frequencies due to genetic drift can be approximated by a

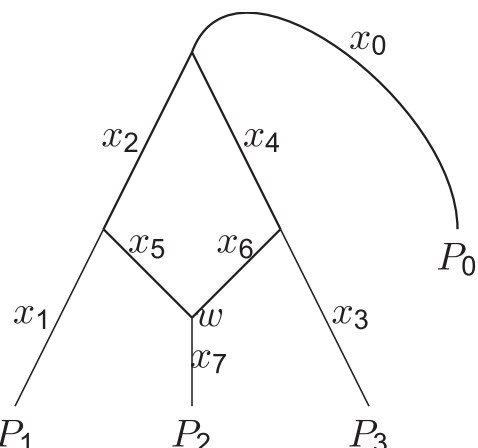

**Fig 4. An admixture graph for the 3 populations and one outgroup.** Considering a single SNP, the quantities $x_1, \ldots,$ $x_7$ are changes in allele frequency, $w$ is the admixture proportion, and $P_0$, $P_1$, $P_2$ and $P_3$ are allele frequencies in the sampled populations. Note that the edge to the outgroup (labeled with $x_0$) is not given a direction. This is because the Gaussian drift model is reversible, meaning that the population split between the outgroup and the other populations could have happened at any point along this branch and identical allelic covariance matrices would be produced. For simplicity, we model the outgroup as the parent of the root node, as described in Method Overview.

normal distribution when the allele frequency change is small and the frequency is far from the boundaries at 0 and 1. If $x_i$ is the amount of drift from a node with allele frequency $p_i$, then the allele frequency change can be approximated as $x_i \sim N(0, (1 - e^{-d_i})p_i(1 - p_i))$ where $d_i = t_i/2N_i$ is the number of generations scaled with the population size [6]. We collect the factor $(1 - e^{-d_i})$ into a single factor $c_i$ and substitute the node-specific $p_i$ with a SNP-global $p$ giving the tractable, approximate, expression

$$x_i \sim N(0, c_i p(1 - p)).$$

Consequently, we can approximate the joint distribution of allele frequencies at all leaf nodes as

$$\begin{pmatrix} P_1 - P_0 \\ P_2 - P_0 \\ P_3 - P_0 \end{pmatrix} \sim\approx N(0, p(1 - p)\Sigma), \ \Sigma = A \cdot \mathrm{diag}(c_0, \ldots, c_7) \cdot A^* \tag{2}$$

where matrix $\Sigma$ is called the *admixture graph covariance matrix*.

The empirical estimate of the covariance of allele frequencies is denoted the *data covariance matrix*. In real data we never observe the population allele frequencies but rather the sample allele frequencies. This complicates the computation of the data covariance matrix slightly. Let $p_{ij}$ be the sample allele frequency in the $i$'th population at the $j$'th SNP, $i = 0, 1, \ldots, n, j = 1, \ldots,$ $N$. They are assumed to come from the distribution

$$p_{ij} \sim \frac{1}{m_{ij}} \mathrm{Bin}(m_{ij}, P_{ij}) \tag{3}$$

where $m_{ij}$ is the number of haplotypes sampled and $P_{ij}$ is the population allele frequency.

Denote population $i = 0$ an outgroup, and consider the intuitive estimate of the covariance matrix

$$S_{k,l} = \frac{1}{N} \sum_{j=1}^{N} (p_{kj} - p_{0j})(p_{lj} - p_{0j})$$ (4)

If there are any missing values in a summand, we leave that summand out of the sum. Regardless of missing values, (4) is inherently biased because the inner term $(p_{kj} - p_{0j})(p_{lj} - p_{0j})$ does not have the same mean as $(P_{kj} - P_{0j})(P_{lj} - P_{0j})$. From (3) we calculate the difference as

$$\mathbb{1}_{\{k=l\}} \frac{P_{kj}(1 - P_{kj})}{m_{ij}} + \frac{P_{0j}(1 - P_{0j})}{m_{ij}}$$

which suggests the following bias correction term for $S_{k,l}$:

$$\hat{B}_{kl} = \mathbb{1}_{\{k=l\}} \frac{1}{N} \sum_{j=1}^{N} \frac{p_{kj}(1 - p_{kj})}{m_{ij} - 1} + \frac{1}{N} \sum_{j=1}^{N} \frac{p_{0j}(1 - p_{0j})}{m_{ij} - 1}.$$

After correcting, we normalize with

$$\hat{h} = \frac{1}{N} \sum_{j=1}^{N} \bar{p}_j (1 - \bar{p}_j), \text{ where } \bar{p}_j = \frac{1}{n+1} \sum_{i=0}^{n} p_{ij}$$

to take the factor $p(1 - p)$ from (2) into account.

If the sample allele frequencies were normally distributed and independent across markers, the estimator in (4) would be Wishart distributed and the degrees of freedom would be the number of markers. The sample allele frequencies are not independent and only approximately normal, yet we use the likelihood

$$W(S/\hat{h}; \Sigma + \hat{B}/\hat{h}, \text{df}).$$ (5)

The degrees of freedom, df, is adjusted to take into account the lack of independence. We estimate df using $R$ bootstrapped replicates of $S/\hat{h}$ which we will denote $X^{(1)}, \ldots, X^{(R)}$. Let $\bar{X}$ be the average of the bootstrap samples. It would be natural to estimate the df with the maximum likelihood of the model

$$X^{(1)}, \ldots, X^{(R)} \sim W(\bar{X}, \text{df})$$ (6)

However, simulations show that the estimates of df from (6) give results that are less accurate than the following moment-based estimator (see S14 Fig). We take advantage of the fact that the variance of the $(k, l)$'th entry of a Wishart distribution with mean $\Psi/\text{df}$ and degrees of freedom, df, is

$$\frac{1}{\text{df}} \left( \Psi_{kl}^2 + \Psi_{kk} \Psi_{ll} \right)$$

to estimate the df as

$$\arg\min_{\text{df}} \sum_{k=1}^{n} \sum_{l=1}^{n} \left( \widehat{\text{Var}}(X_{kl}^{(1)}, \ldots, X_{kl}^{(R)}) - \frac{1}{\text{df}} \left( \bar{X}_{kl}^2 + \bar{X}_{kk} \bar{X}_{ll} \right) \right)^2$$ (7)

where $\widehat{\text{Var}}$ is the sample variance. This moment-based estimator leads to better performance of AdmixtureBayes (S14 Fig).

In practice, to make the inference more robust to deviations from the prior, we normalize the matrices by using the likelihood

$$W(c_S S/\hat{h}; c_S(\Sigma + \hat{B}/\hat{h}), \mathrm{df}) \tag{8}$$

where $c_S = (\log_2(L)L + L)/\mathrm{tr}(S/\hat{h})$. For more on this, see the section "Robustness correction" in S1 Text.

## Prior

We define a prior on the topology, $\mathcal{G}$, and on the continuous parameters of the admixture graph. The continuous parameters include the branch lengths, $\vec{c} = (c_1, \ldots, c_D)$, and the admixture proportions $\vec{w} = (w_1, \ldots, w_K)$. Let $K$ denote the number of admixture events, $L$ the number of leaves, and $D = 2L - 2 + 3K$ the number of branches. The full prior is then

$$P(\mathcal{G}, \vec{c}, \vec{w}) = P(\mathcal{G}|K)P(K)P(\vec{c}|K)P(\vec{w}|K).$$

The prior on the number of admixture events is a geometric distribution with parameter 0.5 (truncated to max 20). The prior on $\mathcal{G}$, $P(\mathcal{G}|K)$, is a uniform prior on all labeled admixture graphs with $K$ admixture events. To evaluate this prior, we need to calculate the number of possible topologies for a given number of admixture events. Therefore we have derived the recurrence formula

$$
\begin{aligned}
N(L, P, K, E) \ = \ & 2(E+1)N(L-1, P, K, E+1) \\
+ \ & (L - 2P + 1)N(L-1, P-1, K, E) \\
+ \ & (L + 2P + 3K - 2E - 2)N(L-1, P, K, E) \\
+ \ & \frac{2(P+1)}{L(L+1)}N(L+1, P+1, K-1, E-1) \\
+ \ & \frac{4(P+1)(P+2)}{L(L+1)}N(L+1, P+2, K-1, E) \\
+ \ & \frac{4(P+1)(L-2P-1)}{L(L+1)}N(L+1, P+1, K-1, E) \\
+ \ & \frac{(L-2P)(L-2P+1)}{L(L+1)}N(L+1, P, K-1, E)
\end{aligned}
$$

where $L$ is the number of leaves, $P$ is the number of pairs of leaves that share a common parent, $K$ is the number of admixture events, $E$ is the number of eyes, and $N(L, P, K, E)$ is the number of unique topologies with those attributes. Notice that we here allow eyes which otherwise are disallowed in our definition of admixture graphs. See S1 Text for proof. Then

$$P(\mathcal{G}|K) = \frac{1}{\sum_{P=0}^{\lfloor L/2 \rfloor} N(L, P, K, 0)}$$

For the admixture proportion prior, $P(\vec{w}|K)$, we chose the uniform distribution on the interval $(0, 1)$.

For the prior on the branch lengths, $P(\vec{c}|K)$, we chose to let all branch lengths be independent and marginally follow the distribution

$$c_i \sim \mathrm{Exp}\left(\frac{2L-2}{D}\right), \ i = 1, \ldots, D \tag{9}$$

The rate of the exponential prior adapts to the topology such that graphs with many branches, and thereby many admixture events, are expected to have smaller branch lengths. For motivation, see the section on Robustness correction in S1 Text.

## MCMC

The MCMC is implemented as a parallel Metropolis coupled MCMC algorithm [26] [27] to increase the number of jumps between local maxima of the posterior surface. Because admixture graphs with different number of admixture events also have different numbers of continuous parameters, we use the reversible jump generalization of the MCMC algorithm [28]. The proposal distribution is a mix of 7 smaller proposals. They are

1. Add an admixture branch to the admixture graph. An admixture branch goes from a *source branch* to a *sink branch* (Fig 5). To make the proposal, a random sink branch, $s$, is chosen with probability $\frac{1}{D}$ where $D$ is the number of branches in the graph (not including the branch to the outgroup). Next, a random source branch, $s'$, is chosen from the remaining branches (including the root/outgroup branch) such that an addition of an admixture branch would not create a cycle in the graph. If the number of possible sink branches is $D'(s)$, the probability of the sink position is $\frac{1}{D'(s)}$. Next the attachment point on the sink branch is simulated uniformly. If the branch lengths of $s$ and $s'$ is $c(s)$ and $c(s')$ the attachment outcome has density $\frac{1}{c(s)c(s')}$. If the source branch is the root branch, we simulate the attachment point with an exponential distribution, Exp(1), instead. The new admixture proportion is simulated uniformly between 0 and 1, and the admixture branch length, $\tilde{s}$, is simulated from Exp(1) with density $e^{-\tilde{s}}$. Lastly, the labeling of the two parent branches of the new admixture node is simulated. The probability of either possible labeling is $\frac{1}{2}$. In conclusion, the density is

$$\frac{1}{D}\frac{1}{D'(s)}\frac{1}{c(s)}\frac{1}{c(s')}e^{-\tilde{s}}\frac{1}{2} \tag{10}$$

   To find the acceptance probability of this proposal, we calculate the proposal probability of the reverse move (see proposal number 2). The reversible jump Jacobian factor is 1.

2. Remove an admixture branch from the admixture graph. An admixture branch can be removed if 1) its parent is not an admixture node and 2) its removal will not cause an eye. Let the number of admixture branches eligible for removal be $K'$. We choose uniformly

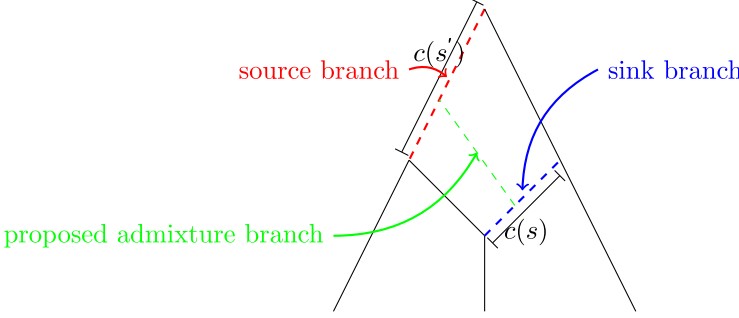

**Fig 5. When adding an admixture branch (green), we will randomly draw the branch where it comes from, the source branch (red).** The admixture branch goes into the sink branch (blue).

from that set and remove the admixture branch. The density is

$$\frac{1}{K'} \tag{11}$$

3. Node sliding. A random branch whose parent is a divergence node is chosen. We move its attachment point to its source branch a distance $\lambda x$ where $x \sim \chi^2(1)$. A node can often be slid either up and down and sometimes the sliding node meets a bifurcation where it can slide in either of two directions. We choose the new node position uniformly from the set of the possible sliding destinations, following the topological constraints defined in step 1. If the sliding node slides out of the graph, we reject the proposal. The forward density is $\chi^2\left(\frac{x}{\lambda}\right)/\omega$ where $\omega$ is the number of possible sliding destinations for a node when moved a distance $x$ from its position in the current graph. We compute the backward density using the same formula. We update $\lambda$ on-the-fly following guidelines for adaptive proposals in MCMC [29], eliminating the need for pre-analysis parameter tuning.

4. Random walk on the branch lengths. We add a normally distributed noise to all the branch lengths. If any branch length becomes negative, we take its absolute value. This is known as a reflecting boundary random walk. The backward density is identical to the forward density. The variance of the random walk increments is controlled by parameter $s$ which we also adapt on-the-fly using adaptive strategies.

5. Resample admixture proportions. We sample each admixture proportion in the graph independently from the standard uniform distribution.

6. Random walk on the branch to the outgroup as in step 4 but with another $s$-value. Negative proposed branch lengths are again reflected.

7. Random walk on the branch lengths but inside the null space of matrix $A$. This means that the proposed admixture graph will have the same covariance matrix—and therefore the same likelihood—as the previous graph. This proposal is also adaptive, as in step 4.

## Graph summaries

In the Results section we explained the two summaries, *minimal topology* and *consensus graph*, which we will define formally here. Furthermore, we introduced the Set Distance used to measure distances between admixture graph topologies and the Covariance Distance for distances between admixture graphs. In this section, we define these quantities.

The Covariance Distance between two admixture graphs with $L$ leaves and covariance matrices $\Sigma$ and $\Sigma'$, respectively, is

$$\sqrt{\sum_{i=1}^{L}\sum_{j=1}^{L}\left(\Sigma_{ij} - \Sigma'_{ij}\right)^2} \tag{12}$$

For a single node let the *descendant set* be the the set of its leaf descendants, e.g. $t = \{l_1, l_2, \ldots, l_a\}$. For a topology, let $T$ be the *topology set*, which is the set of descendant sets of all its nodes, except the leaves and the root. This idea of a topology set is similar to the idea in the software package PhyloNet of considering the set of clusters induced by each edge of an evolutionary network [10] [12]. The *minimal topology* is the extension of such a topology set to a

directed graph. The extension starts by adding the trivial descendant sets for the leaves (containing only one leaf) and the root (containing all the leaves). Denote this set $\mathcal{T}$. The minimal topology has a node for each element of $\mathcal{T}$ and there is a connection from node $t \in \mathcal{T}$ to $t' \in \mathcal{T}$ if

$$t \neq t' \tag{13}$$

$$t' \subseteq t \tag{14}$$

and

$$\nexists\, t'' \in \mathcal{T} \setminus \{t, t'\} : t' \subseteq t'' \subseteq t \tag{15}$$

To summarize a sample of admixture graphs, $g_1, \ldots, g_R$, using a consensus graph, we first transform all of them into their topology sets and obtain a sample $T_1, \ldots, T_R$. The posterior probability of a node can be estimated by the sample frequency

$$f(t) = \frac{|\{T \in \{T_1, \ldots, T_R\} : t \in T\}|}{R}$$

The topology set of the *consensus graph* at threshold $\alpha$ is

$$T^\alpha = \left\{ t \in \bigcup_{i=1}^{R} T_i : f(t) > \alpha \right\}. \tag{16}$$

The consensus graph itself is obtained by extending $T^\alpha$ to a directed graph with the rules (13)–(15).

The *Set Distance* between two graphs $g_1$ and $g_2$ with topology sets $T_1$ and $T_2$ is

$$|T_1 \setminus T_2| + |T_2 \setminus T_1| \tag{17}$$

## Running AdmixtureBayes, TreeMix, and OrientAGraph

As mentioned in the Results section, we simulated 20 datasets each from 4 distinct admixture graph models, and ran AdmixtureBayes, TreeMix, and OrientAGraph on each dataset. We compared the accuracy of each method using three metrics described in the previous section: the Topology Equality, Set Distance, and Covariance Distance. Comparing their accuracy is not straightforward because TreeMix and OrientAGraph produce one graph whereas AdmixtureBayes produces a posterior sample of graphs. In addition, TreeMix and OrientAGraph assume a fixed number of admixture events, whereas AdmixtureBayes samples graphs with different numbers of admixture events. TreeMix and OrientAGraph can estimate a maximum likelihood graph for a fixed number of admixture events, but the higher the number of admixture events, the higher the maximum likelihood value. Therefore, the original TreeMix paper suggests iteratively adding admixture events and stopping when the added admixture event does not pass a test for statistical significance. However, to simplify the comparison, we ran TreeMix and OrientAGraph with the true number of admixture events. We considered all graphs produced by AdmixtureBayes, even those with the wrong number of admixture events. We note that this could increase the error of AdmixtureBayes.

TreeMix and OrientAGraph first estimate an initial admixture-free tree by iteratively adding best fitting populations in a random procedure. Next, the admixture branches are added

deterministically (although the exact method for adding branches differs between the two methods). Because of the randomness of the first step, the starting seed could influence the results. However, preliminary results showed that repeating the TreeMix maximum likelihood optimization for different seeds and choosing the highest likelihood graph amongst the repeated analyses did not change the accuracy of the estimated admixture graphs when analyzing our simulated datasets. Most seeds produced the same maximum likelihood graphs, an observation also found by Molloy *et al*. [3]. Therefore, we used only one seed for both TreeMix and OrientAGraph.

AdmixtureBayes was run on 20 datasets generated from msprime [18] from 4 distinct admixture graphs (see Fig 1). The Mode graph was chosen as the graph with the highest posterior out of all sampled graphs. Then, the first 35% of each chain was discarded as a burn-in and 100 equally spaced graphs were sampled from the resulting collection of graphs for use in the AdmixtureBayes Mean estimates. The exact code for running each of these analyses is in the SimulationStudy folder on the AdmixtureBayes GitHub.

## Data

We analyzed a dataset consisting of SNPs for 12 human populations that was first analyzed by Moreno-Mayer *et al*. [17]. We treated the Yoruba population as an outgroup leaving effectively 11 populations with unknown relationships to estimate. One diploid individual was sampled from each population, except the Koryak, Ket, Greenlander and Athabascan populations, which each had two diploid individuals. Whole genome-sequencing was performed on each individual to provide an average coverage between 1X (for the Malta individual) to 44.2X (for one of the Greenlander individuals). Further details regarding sequencing and data processing methods are described in Moreno-Mayer *et al*. [17]. The alleles for the ancient individuals from the populations Saqqaq, Malta, Anzick and USR1 that were not transversions were treated as missing. We then filtered out any site for which there was a population with missing data. In total 251,542 biallelic SNPs were retained. Large numbers of missing SNPs for some individuals is not a computational problem for AdmixtureBayes, though it does violate the assumption of even sampling imposed by the Wishart distribution (see Methods, Eq (5)). Both the original VCF file and the allele count file used as input to AdmixtureBayes are available on the AdmixtureBayes GitHub.

## Supporting information

**S1 Fig. In all of our illustrations the direction of edges is from top to bottom, unless marked otherwise.** This multigraph topology has 4 leaves, 1 pair, 2 admixture nodes, 5 divergence nodes, and 1 eye. The root is the node at the very top of the topology. We have not explicitly labeled the nodes and branches.
(EPS)

**S2 Fig. Representatives of the sets $\mathcal{T}_{3,1,1,0}$ (left), $\mathcal{U}_{3,1,1,0}$ (center) and $\mathcal{S}_{3,1,1,0}$ (right).** In all of our illustrations on labeled or unlabeled topologies, the admixture edges in $\mathcal{E}_A$ are marked with a dashed line. Here $|\mathcal{S}_{3,1,1,0}| = 2$, $|\mathcal{U}_{3,1,1,0}| = 4$ and $|\mathcal{T}_{3,1,1,0}| = N(3, 1, 1, 0) = 12$.
(EPS)

**S3 Fig. Illustration of a shape, $S_1$ (left), and the three unlabeled topologies corresponding to a shape $S_2$ ($S_2$ not explicitly drawn).** We have $S_1, S_2 \in \mathcal{S}_{2,0,3,1}$, $|\mathcal{U}_{S_1}| = 2$ and $\mathcal{U}_{S_2} = \{U_1, U_2, U_3\}$. Furthermore, the leaves of $U_1$ and $U_2$ are indistinguishable, while the leaves of $U_3$ can be told apart. To see this, follow the path from the leaves to root; in $U_1$ and $U_2$ the path will only depend on whether the parent branch of the first encountered admixture is

in $\mathcal{E}_M$ or $\mathcal{E}_A$ and not on the starting leaf. In contrast the starting leaf does matter for $U_3$ so the leaves are distinguishable. Hence, $|\mathcal{T}_{U_1}| = |\mathcal{T}_{U_2}| = 1$ but $|\mathcal{T}_{U_3}| = 2$.
(EPS)

**S4 Fig. The two shapes of $\mathcal{S}(5, 2, 0, 0)$, denoted $S_1$ and $S_2$ are illustrated above.** Here, $\mathcal{U}_{S_1} = \{U_1\}$ and $\mathcal{U}_{S_2} = \{U_2\}$, because there are no admixture edges. Interestingly, the shape $S_2$ exhibits more symmetry than the shape $S_1$. To see this, let $G_1$ and $G_2$ be representatives of $U_1$ and $U_2$ with leaves labeled $l_1, l_2, l_3, l_4, l_5$ from left to right. In both cases $|\mathcal{T}'_{U_1}| = |\mathcal{T}'_{U_2}| = 5! = 120$. The group $H_{G_1} = \langle e, (12), (34) \rangle$ has four elements and so $|\mathcal{T}_{U_1}| = 120/4 = 30$. The group $H_{G_2} = \langle e, (12), (34), (13)(24) \rangle$ has eight elements and so $|\mathcal{T}_{U_2}| = 120/8 = 15$. Altogether, $N(5, 2, 0, 0) = 15 + 30 = 45$. Notice that the leaves $l_4$ and $l_5$ form a pair in one fifth of the elements in both $|\mathcal{T}_{U_1}|$ and $|\mathcal{T}_{U_2}|$ although the two sets are of different size.
(EPS)

**S5 Fig. Example graphs and their predecessors from each sub case 1.1)—2.4).** The graph $\rho$ $(G_{1.2})$ is the only labeled admixture graph that doesn't have a predecessor, and the ultimate predecessor of every other graph.
(EPS)

**S6 Fig. From the posterior AdmixtureBayes samples, we computed the posterior probability of all nodes.** The above graph is the smallest directed graph with all the nodes that have a posterior probability higher than 75%. Each internal node is colored according to its posterior probability, as described in Fig 3.
(EPS)

**S7 Fig. From the posterior AdmixtureBayes sample, we computed the posterior probability of all minimal topologies for several subsets of the populations.** Here we show the three topologies with the highest posterior. The listed posterior for each graph represents the percentage of sampled graphs that have this minimal topology induced by the relevant set of nodes. For example, in 96% of graphs sampled, if only the leaves Athabascan, Koryak, and Saqqaq are considered, then all non-root and non-leaf nodes have a topology set of {Athabascan, Koryak}. The percentages in each node are the percentage of sampled graphs that have a node with the topology set implied by that node. For example, in 96% of graphs sampled, if only the leaves Athabascan, Koryak, Saqqaq are considered, then there is at least one node with the topology set {Athabascan, Koryak}. Whether a graph with this node belongs in the top left or bottom left box will depend on the presence or absence of a node with the topology set {Athabascan, Saqqaq}, which happens in 1% of all sampled graphs.
(EPS)

**S8 Fig. Continuation of S7 Fig.**
(EPS)

**S9 Fig.** The method used to calculate the Set Distance between two admixture graph topologies (left). First, the topologies are transformed in their descendant sets/topology sets (middle). The distance is then calculated as the symmetric set distance between the two topology sets (right).
(TIF)

**S10 Fig. Examples of how the minimal topology is calculated.** First, we derive the topology set (middle) from the topology (left). The minimal topology (right) is the smallest possible graph that is consistent with the topology set. Note, node labels assigned to the topology (left)

are arbitrary and do not identify corresponding nodes in the minimal topology (right).
(TIF)

**S11 Fig. Here, we plot the trace plots for our simulated dataset.** Each chain is shown as a separate column. Each summary statistic is shown as a separate row.
(EPS)

**S12 Fig. We plot the Gelman-Rubin convergence diagnostics on our simulated dataset for our three summary statistics after a burn-in fraction of 0.35.** A rapid convergence to 1 indicates that this is a sufficient burn-in period.
(EPS)

**S13 Fig. We here show the autocorrelation plots for the summary statistics of our simulated data after a burn-in fraction of 0.35.** We only show the results for Chain 1 and do not include the number of admixture events as the autocorrelation shows strange behavior for discrete variables.
(EPS)

**S14 Fig. We simulated admixture graphs with 10 leaves and 0, 1 and 2 admixture events.** Using these graphs, we simulated datasets using ms with different sample sizes. The top plot illustrates the ratio between the maximum likelihood degrees of freedom estimate from Eq (6) and the variance estimator in Eq (7). We ran AdmixtureBayes with the maximum likelihood estimate (MLE), the variance estimate (VAR), and 2 and 4 times the variance estimate (VARx2 and VARx4 respectively). We calculated the Mean Topology Equality, which was maximized when using the VAR estimates.
(EPS)

**S15 Fig. We here plot the results of our simulations evaluating the effect of sampling small numbers of haplotypes.** Each boxplot represents the samples of posterior probabilities obtained by running AdmixtureBayes on 100 different datasets, each simulated from the same model with the underlying population history ((pop1,pop2),pop3). We plot a horizontal dashed line at 1/3, which would represent equal posterior probability at each topology. AdmixtureBayes does sample some admixture graphs that do have admixture events, but the 3 non-admixed topologies we list here account for more than 99% of the sampled admixture graphs in each case, so we simply ignore sampled topologies that do have admixture events. We see that increasing the number of sampled haplotypes causes the posterior probability to concentrate on the true graph, while AdmixtureBayes correctly models the uncertainty inherent to sampling fewer haplotypes.
(EPS)

**S16 Fig. Here we plot the results obtained when running TreeMix and OrientAGraph on our dataset of Native American and Arctic populations.** We run each method with 3, 4, and 5 admixture events as these were the numbers of admixture events in nearly all graphs sampled by AdmixtureBayes after the burn-in period (see S11 Fig).
(TIF)

**S1 Text. Supplementary information.**
(PDF)

## Acknowledgments

We thank the members of the Nielsen Lab for their helpful thoughts on both the manuscript and the AdmixtureBayes GitHub documentation. We also thank J. Víctor Moreno-Mayar for his helpful discussions on analyzing the real dataset.

## Author Contributions

**Conceptualization:** Thomas Mailund, Rasmus Nielsen.

**Formal analysis:** Svend V. Nielsen, Andrew H. Vaughn.

**Investigation:** Svend V. Nielsen, Andrew H. Vaughn.

**Methodology:** Svend V. Nielsen, Kalle Leppälä, Michael J. Landis, Rasmus Nielsen.

**Software:** Svend V. Nielsen, Andrew H. Vaughn.

**Supervision:** Thomas Mailund, Rasmus Nielsen.

**Validation:** Andrew H. Vaughn.

**Visualization:** Svend V. Nielsen, Andrew H. Vaughn, Kalle Leppälä.

**Writing – original draft:** Svend V. Nielsen, Andrew H. Vaughn, Kalle Leppälä,

Rasmus Nielsen.

**Writing – review & editing:** Andrew H. Vaughn, Michael J. Landis, Rasmus Nielsen.

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
