## [Decision Letter · Decision Letter 0]

7 Nov 2022

Dear Dr Vaughn,

Thank you very much for submitting your Methods entitled 'Bayesian inference of admixture graphs on Native American and Arctic populations' to PLOS Genetics.

The manuscript was fully evaluated at the editorial level and by independent peer reviewers. The reviewers appreciated the attention to an important problem, but raised some substantial concerns about the current manuscript. Based on the reviews, we will not be able to accept this version of the manuscript, but we would be willing to review a much-revised version. We cannot, of course, promise publication at that time.

If you decide to revise the manuscript for further consideration at PLOS Genetics, please aim to resubmit within the next 60 days, unless it will take extra time to address the concerns of the reviewers, in which case we would appreciate an expected resubmission date by email to plosgenetics@plos.org.

We are sorry that we cannot be more positive about your manuscript at this stage. Please do not hesitate to contact us if you have any concerns or questions.

Yours sincerely,

Sharon R. Browning

Guest Editor

PLOS Genetics

David Balding

Section Editor

PLOS Genetics

The reviewers were all positive about this manuscript, but have suggestions for improving the work. We invite the authors to address as many of these suggestions as possible in a revised submission.  In preparing your revision please consider using the PLOS Genetics format for Methods papers, see https://journals.plos.org/plosgenetics/s/submission-guidelines#loc-manuscript-organization

Reviewer's **Comments to the Authors:**

Reviewer #1: See attached PDF.

Reviewer #2: S. Nielsen et al. have developed a novel method for characterizing admixture graphs. The authors adapt an existing allele frequency covariance approach and add MCMC statistical machinery that allows a more rich characterization of admixture graphs compared to existing methods that mostly focus on estimating a single best, admixture graph.

I agree that searching for the best topology, or a set of reasonable ones, for an admixture graph is a real problem, as existing methods are not particularly well suited to finding the set of high-probability admixture graphs. These methods often require extensive direct input and subjective judgment, making them difficult to apply in a consistent manner. The authors also present a small suite of topological-based methods for summarizing admixture graphs that are potentially useful moving forward.

The authors apply the new method to simulated data, compare the results with an existing related method, Treemix, and apply the method to a set of 11 population samples with a goal of better understanding the genetic relationship between Koryak, Saqqaq, Athabascans, and native Greenlanders.

It is a substantial amount of work to develop a MCMC procedure to efficiently sample the huge space of possible admixture graphs. The authors present a set of 7 proposals for alterations to graphs that seem to be able to move between possible graphs in a way that respects detailed balance. It is difficult to evaluate the extent that the full space of possible graphs is explored, but I will assume it is. Towards this goal, the authors use an adaptive Metropolis-Coupled MCMC that attempts to better explore multimodal distributions.

While I found the text generally clear, I found the structure and flow of the manuscript challenging. PLOS Genetics prescribes a Results/Discussion/Methods format. I understand this structure has the potential to make it difficult to find the best way to present Methods-focused papers, but I think the manuscript would benefit from reorganization.

For example, the results section starts with an evaluation of MCMC chains, while this is an important aspect of conducting any MCMC analysis, it is not really the main focus of the current study, and no results are actually presented in the text. It is followed by a comparison to Treemix section that spends 1.5 pages describing the methods for the comparison. Again this is important, but should not precede the basic description of AdmixtureBayes. Main figures 3, 4, 5, 6, 7 are only referenced in the appendix, and all figures applying the method to data (simulated or real) are supplemental figures. This is out of step with the title that highlights applying this method to real data. “... on Native American and Arctic populations”.

I suggest the Results section should cover the following topics in order - 1) Method introduction, justification and important details, 2) Evaluation on simulated data / comparison to other methods 3) Application to real data. This would better match other papers in the field, such as the Treemix paper, also published in PLOS Genetics. This also better matches the alternative formatting for Methods papers at: https://journals.plos.org/plosgenetics/s/submission-guidelines.

The Methods section can supplement this formatting, by starting with e.g., 1) a full description of the method, 2) Description of simulations / comparisons to other methods, 3) real data sets. Self-contained details of specific aspects can be moved to an appendix or supplement.

I was able to download and install AdmixtureBayes from the provided github link and to run an analysis on the provided example data file without any problems. My strong suggestion for the software would be to utilize a common data format, such as VCF, rather than a file format unique to a particular program. Also I might suggest splitting step 1 in the analysis outline into two steps, 1) estimation of the population covariance matrix and 2) run the MCMC. I did not see any built-in way to estimate or examine the covariance matrix.

Small issues:

The authors show how small numbers of sampled haplotypes can reduce the accuracy of estimated admixture graph topology (figure S2). However, it was not immediately clear if AdmixtureBayes results accurately characterize the uncertainty due to sampling few haplotypes. This seems crucial to interpreting the results applying AdmixtureBayes to real data, such as in the current manuscript.

Due to a lack of background, I was not able to adequately review the topological arguments present in Appendix A.3, (the justification of the flat prior on the topology). This could be split off into a separate manuscript to be submitted to another journal such as Theoretical Population Biology, or another appropriate journal where it could stand on its own and get a separate review.

The application of AdmixtureBayes to the real data set and the accompanying discussion did a good job highlighting how often researchers have specific questions, such as “is there admixture present?”, “what is the source of admixture in this population?”, etc. As a suggestion it could be useful to examine different types of relationships between populations, and how well they are characterized by AdmixtureBayes. Currently this is all done in figure S3, using the mode set distance statistic, which is not so easy to see how to interpret biologically.

It was difficult to find the number of simulation and analysis replicates that the simulation and Treemix comparison results were based on.

The code for the two simulation evaluations are nice to include in the manuscript, but a higher level explanation of the demography, as well as a statement of why this is an appropriate demography / admixture graph would aid understanding of the results.

Page 4- “It has previously been used in several other methods aimed at modeling the joint distribution of allele frequencies among populations” please support with citations.

Page 7 - “We then ran AdmixtureBayes for three different chains, each with --MCMC chains 16” is this 3, 16 or 48 chains?

Page 19 - Admixture graphs are described here, and this could be useful if the rest of the paper utilizes this description. However, it is not clear how to match the description here with the example admixture graph present in Figure 1. This description states “There exists one and only one root. That is a node with no parents and exactly two children”. Yet, to my eye, Figure 1 contains zero nodes with no parents and two children.

Page 35 - It would help to provide a figure of the admixture graph that is simulated by this code.

Figures

In general I found the figures under-described, with many explanations not repeated on each relevant figure. Many main figures were not referenced in the main text, but I appreciated the figure representing topological concepts. In general, I thought the supplemental figure were better suited for main, while most main figure were better suited to the supplement.

S1 - Should be moved to main. Reorder the subplots so similar statistics are next to each other - eg . (mode, mean) Topology equality, (mode mean) set distance.

S3 - why was only this statistic out of the 4 based on topology selected to be shown.

S4 - why are the two subfigures different sizes? Node labeling could be improved as it is difficult to distinguish the support from the node label. Please explain coloring in the figure legend.

S6 and S7 I think these plots are good examples of the additional utility possible with AdmixtureBayes. However in this presentation I did not understand how the posterior of the interior nodes combined with the posterior for the topology e.g. the ((Koryak, Saqqaq), Athabascan) tree.

Reviewer #3: Manuscript Review

PGENETICS-D-22-01018

“Bayesian inference of admixture graphs on Native American and Arctic populations"

In this study, the authors evaluated the ability of admixture graphs to reconstruct population relationship in the Americas based on genome data. They present a new reversible jump MCMC algorithm for sampling high-probability admixture graphs and show that this approach works well both as a heuristic search for a single best-fitting graph and for summarizing shared features extracted from posterior samples of graphs. They subsequently used this method with data from 11 Native American and Siberian populations to address the relationship between Saqqaq, Inuit, Koryaks, and Athabascans. In contrast with previous work, they find that “the Saqqaq is not a good proxy for the previously identified gene flow from Arctic people into the Na-Dene speaking Athabascans.”

From my assessment of the admixture graph analysis, the authors’ analysis seems robust, and has been directed at issues concerning optimality and likelihood issues arising from the use of other admixture graphing methods commonly employed in population genetics studies.

Yet, having followed the development of admixture graphs over a number of years, I would like to pose some questions to the authors about their use for modeling the peopling of the Americas based on ancient and modern genome data.

From an admixture standpoint, how accurately can a single genome represent a “population”? Can the Saqqaq individual really represent an entire ancestral group? What is the consequence of adding more individuals per population when estimating admixture graphs? Can this effect be modeled in order to determine how population level data (e.g., 20-30 individuals) will affect the accuracy of these estimates? How much certainty can we place on the results of admixture analysis with ancient and modern individuals when data from key ancient populations are lacking, i.e., those ancestral to modern groups which now occupy certain regions of northeast Asia or North America?

These are germane questions for efforts to reconstruct the peopling of the Americas, when initial population movements into North America may have been pulsatile in nature and those occurring in Beringia during the latter stages of the LGM may have been quite dynamic.

On this same note, modeling the relationships between circumarcric populations is not necessarily easy. Greenlanders are not precisely the same group as Alaskan Inuit, Yupik, Aluttiq or Aleut populations based on language and culture, which took hundreds if not thousands of years to evolve. The same thing is true to some extent with genetic data, which suggests subtle differences between Eskimoan-speaking groups and more substantial ones between these groups and Aleuts and Athapaskans (Na-Dene). The Athapaskans used in this study also represent one only group of Na-Dene speakers whose homeland may be Southeast Alaska and from which they spread into what is now Alaska and Canada.

In this regard, the authors seem to indicate that they consider Athapaskans to be “Native Americans” inclusive of Amerindians (the first wave of settlers). Is this the case or is the term “Native American” being used here simply to distinguish between Eskaleut-speaking groups and all other indigenous population of the Americas?

I do find it intriguing that the authors’ analysis suggests that that Athapaskans are best represented as “the result of admixture between a Native American population and a Siberian population most closely related to the Koryak, but not the Saqqaq.” However, previous studies have produced Y-chromosome data linking circumarctic groups including the Saqqaq and Koryaks, and the Saqqaq also show affinities with the Nganasan, Koryak, and Chukchi of northeastern Siberia based on autosomal data. In light of these findings, how can we explain the admixture results presented by the authors?

While recognizing that this manuscript has been submitted to a genetics paper, it is important to reconcile genetic admixture studies such as this with other archaeological, ethnographic, genetic and linguistic evidence to ‘ground truth’ the results of the modeling work. For this reason, I would be most interested to know the authors’ perspective on these issues.

Reviewer #4: Here, Nielsen et al (2021) present a Bayesian method for estimating admixture graphs from allele frequency data. Their approach AdmixtureBayes is evaluated on simulated and real data sets. The results on simulated data sets are promising, and the results on biological data (on "peopling of the Americas") is quite interesting as well (although I am not a domain expert). Overall, this paper makes a strong contribution to the literature on admixture graphs, and I expect the AdmixtureGraph methodology to be of interest to method developers and method users alike.

Nevertheless, this paper could be improved in several ways. I make the following recommendations.

(1) The biological data set should be made publicly available (specifically the authors should provide the VCF and allele frequencies rather than just the accession IDs for genomes).

(2) The comparison study should include an additional method (their current comparison is against TreeMix but a new method OrientAGraph has been shown to improve the topological accuracy of TreeMix).

(3) There are some figures that should be updated so that readers can more quickly understand the differences between running TreeMix and AdmixtureGraph on simulated and biological data sets.

Details are below.

#1. Data Availability

---

This is a major area of research with new methods being developed each year. A limiting factor to comparing methods and developing new methods is that many studies do not publish their data in a usable form. Often only acession IDs for genomes are provided, even though the developed methods require VCFs or allele frequencies as input.

+1a. It looks like Nielsen et al (2021) currently only provide a link to the original Nature paper (which in turn only provides accession IDs). I strongly recommend that Nielsen et al (2021) to make their processed data (i.e., VCF and allele frequencies) publically available. This will make their results reprocible and will help future studies, increasing the impact of this paper. (of all of my comments, this is the one that I feel is the most important to address)

+1b. It's great that Nielsen et al (2021) make the msprime commands available in the paper so that future researchers can benchmark methods on the simulated data sets related to this study (they go above and beyond and provide the ms command as well). I encourage the authors to upload the simulated data sets themselves to Github or some other platform.

#2. Discussion of and comparison to existing methods

---

+2a. Nielsen et al (2021) discuss existing admixture graphs methods on pages 3-4, focusing on well-established methods, like TreeMix (2012), qpGraph (2012), and MixMapper (2013). I recommend discussing some more recently developed methods, like miqoGraph (2021) and OrientAGraph (2021), in addition.

+2b. Nielsen et al (2021) write on page 4 "TreeMix searches through many potential admixture graphs without user input." I recommend changing this sentence to read something like "TreeMix searches through potential admixture graphs without user input by way of an efficient greedy heuristic." This would tie into the comments on page 4 about the downsides of greedy algorithms (where they fail to effectively search network space, getting trapped in local optima).

+2c. I recommend that Nielsen et al (2021) include the command used to run TreeMix on page 37ish. It could also be interesting to note that their observation (that changing the seed doesn't impact the graph topology recovered by TreeMix very much) was also reported by Molloy et al (2021), although these authors varied the population addition order explicitly rather than the seed.

+2d. I recommend that Nielsen et al (2021) add OrientAGraph to their comparison study because OrientAGraph has been demonstrated to produce graphs that are as accurate as those produced by TreeMix or else more accurate. This comparison should be relatively easy to add because OrientAGraph was shown to be negligibly slower than TreeMix on a data set with 10 populations and 2 admixture graphs. Moreover, OrientAGraph takes the same input as TreeMix (allele frequencies) and has nearly the same command. The only difference is the addition of -mlno 1,2 and -allmigs 1,2 flags (where 1,2 indicate that the admixture graph topology search should be expanded with the MLNO and ALLMIG algorithms after adding the first gene flow edge and after adding the second gene flow edge).

+2e. It would be great to add a comparison to miqograph as well but this method is more difficult to run in my experience because it relies on Gurobi and the output of the admixture graph is in a non-standard format.

#3. Admixture Bayes Method

---

+3a. Nielsen et al (2021) write on page 3, "We here improve on these approaches by developing a novel MCMC sampling method, AdmixtureBayes, that can sample from the posterior distribution of admixture graphs. This enables an efficient search of the entire state space a well as the ability to report a level of confidence in the sampled graphs." I recommend replacing "efficient" with "effective" or some other adjective because this study at present doesn't emphasize efficiency. In particular, metrics like runtime are not currently reported for all methods / data sets studied. In addition, the authors do not report the number of admixture graph topologies explored by the different methods. My guess is that the MCMC method is less efficient because the runtime on the biological data set was 50 hours (my guess is that TreeMix would take just a few minutes on this data set discounting the time to estimate summary statistics from allele frequencies).

+3b. I found the description of AdmixtureBayes to be well-written and well-motivated. My only questions was how the MCMC begins. The proposals must be performed on some starting admixture graph (both topology and numerical parameters), so I was curious whether this graph was selected at random or constructed using some heuristic, or something in between.

+3c. Nielsen et al (2021) define the concepts of minimal topology and consensus graphs, along with tools for computing and visualizing them. It would be great could if the authors discuss how these ideas compare to the techniques used by existing methods (like PhyloNet), which I believe also produce summarizes of network space.

#5. Results

---

+5a. I found Figure S1/S2/S3/S14 to be somewhat difficult to interpret because only a single value is reported for multiple data sets. I recommend Nielsen et al (2021) re-plot these figures as boxplots (with data points plotted over the boxes) or as violin plots. It could also be helpful to plot topological error for TreeMix (e.g. set distance between true graph and estimated graph) against error for AdmixtureBayes (e.g., set distance between true graph and MAP graph). This way would cleary show when AdmixtureGraph is outperforming TreeMix for each data set. Alternatively, the difference between methods on a given data set could be plotted as box plot.

+5b. Currently, the authors provide a text description of differences between prior analyses on the "peopling of the Americas" and the results of running AdmixtureBayes. It would be very helpful to show this information as a figure. For example, it would be great if Nielsen et al (2021) provided the result of running TreeMix as part of Figure S4, with the same node labelings.

+5c. I recommend the authors compute likelihood scores for the graphs shown in Figures S4, S5, and S6 and include this information in the figure (it seems like numerical parameters could simply be reoptimized for the given topologies and then the likelihood scores reported). This information would help readers compare the results to prior studies where likelihood scores have been reported. It could also emphasize points being made by Nielsen et al (2021) about the values of Bayesian methods compared to ML methods in this context.

+5d. I recommend Nielsen et al (2021) provide the specific commands used for running TreeMix and AdmixtureBayes be provided (or provide the analysis scripts).

#6. Links:

---

miqograph - https://doi.org/10.1093/bioinformatics/btaa988

OrientAGraph - https://doi.org/10.1093/bioinformatics/btab267

PhyloNet - https://doi.org/10.1093/sysbio/syy015

**Have all data underlying the figures and results presented in the manuscript been provided?**

Reviewer #1: Yes

Reviewer #2: Yes

Reviewer #3: Yes

Reviewer #4: **No: **See my review

PLOS authors have the option to publish the peer review history of their article (what does this mean?). If published, this will include your full peer review and any attached files.

Reviewer #1: No

Reviewer #2: No

Reviewer #3: No

Reviewer #4: No

---

## [Decision Letter · Decision Letter 1]

23 Jan 2023

Dear Dr Vaughn,

We are pleased to inform you that your manuscript entitled "Bayesian inference of admixture graphs on Native American and Arctic populations" has been editorially accepted for publication in PLOS Genetics. Congratulations! In your final submission, please address the four minor points raised by reviewers 2 and 4.

Yours sincerely,

Sharon R. Browning

Guest Editor

PLOS Genetics

David Balding

Section Editor

PLOS Genetics

Comments from the reviewers (if applicable):

Reviewer's **Comments to the Authors:**

Reviewer #1: All of my comments have been thoroughly addressed. I think the paper makes an important and practical methodological contribution, and warrants publication.

Reviewer #2: S. Nielsen and coauthors have substantially updated their manuscript and addressed many of the reviewers first round issues. Specifically, they have reorganized the main and supplemental texts, reworked the analysis of simulated data, compared to a wider array of alternative methods, updated the MCMC proposal, and made a number of smaller changes to the text and computer program.

I have just two very small comments on the revised version.

In Figure 2 the text labels are too small. In addition, in my opinion the scale is off - any moderate differences between methods are squashed by the figure design.

I appreciate that the authors added further investigations of the effect of smaller sample sizes, even adding a new section to address it specifically (Section A.4). However, I disagree with one specific statement the authors make in that section. The authors state: “we observe that while the true topology is indeed the one inferred by Admixture-

Bayes to have the highest posterior in both cases, the simulated datasets with 40

haplotypes generate a probability distribution that is much more concentrated

on the true admixture graph. We therefore conclude that AdmixtureBayes correctly characterizes uncertainty due to sampling small numbers of haplotypes

from populations.”

However the authors never actually evaluate if this uncertainty is accurately quantified, instead they simply observe that it is increased with smaller sample sizes, consistent with what is expected. I suggest that the authors should simply note this consistency and remove the claim that the uncertainty is accurately characterized.

Reviewer #4: I was very positive about the paper during the first round of review, and all of my comments (about method comparisons, evaluation metrics, and supporting information) have been addressed (for reference I was reviewer #4). I found the new results in the revised paper quite interesting. The simulation study shows that AdmixtureBayes is effective in reconstructing admixture scenarios were existing methods fail (note the results of the biological data set are interesting but I am not a domain expert here). My opinion is the the AdmixtureBayes method will be of great interest to may readers of PLOS Genetics.

Minor comments:

---------------

Potential typo on page 6: "1 of these nodes is called the outgroup and has exactly one child, the root." This makes sense to me from the phylogenetics network literature and in the context of the model, but it could be confusing to some readers looking at Figure 1 (where the outgroup is a leaf vertex that has the root as its parent).

Potential typo on page 29: AdmixtureGraph GitHub -> AdmixtureBayes Github?

**Have all data underlying the figures and results presented in the manuscript been provided?**

Reviewer #1: Yes

Reviewer #2: Yes

Reviewer #4: Yes

PLOS authors have the option to publish the peer review history of their article (what does this mean?). If published, this will include your full peer review and any attached files.

Reviewer #1: No

Reviewer #2: No

Reviewer #4: No

**Data Deposition**

http://datadryad.org/submit?journalID=pgenetics&manu=PGENETICS-D-22-01018R1

**Press Queries**

---

## [Editor Report · Acceptance letter]

7 Feb 2023

PGENETICS-D-22-01018R1 

Bayesian inference of admixture graphs on Native American and Arctic populations 

Dear Dr Vaughn, 

We are pleased to inform you that your manuscript entitled "Bayesian inference of admixture graphs on Native American and Arctic populations" has been formally accepted for publication in PLOS Genetics! Your manuscript is now with our production department and you will be notified of the publication date in due course.

With kind regards,

Timea Kemeri-Szekernyes

PLOS Genetics

On behalf of:
